# Predicted Reversal in **N**-Methylazepine/**N**-Methyl-7-azanorcaradiene Equilibrium upon Formation of Their **N**-Oxides

**DOI:** 10.3390/molecules25204767

**Published:** 2020-10-16

**Authors:** René Fournier, Alexa R. Green, Arthur Greenberg, Edward Lee-Ruff, Joel F. Liebman, Anita Rágyanszki

**Affiliations:** 1Department of Chemistry, York University, Keele Campus, Toronto, ON MJ3 1P3, Canada; renef@yorku.ca; 2Department of Chemistry, University of New Hampshire, Durham, NH 03824, USA; ag1211@wildcats.unh.edu; 3Department of Chemistry and Biochemistry, University of Maryland Baltimore County, Baltimore, MD 21250, USA; 4Department of Physics and Astronomy, York University, Keele Campus, Toronto, ON MJ3 1P3, Canada

**Keywords:** **N**-methylazepine, **N**-Methyl-7-azanorcaradiene, **N**-methylpyrrole, amine **N**-oxides, aromaticity, antiaromaticity, DFT calculations, NICS

## Abstract

Density functional calculations and up to five different basis sets have been applied to the exploration of the structural, enthalpy and free energy changes upon conversion of the azepine to the corresponding **N**-oxide. Although it is well known that azepines are typically much more stable than their 7-azanorcaradiene valence isomers, the stabilities are reversed for the corresponding **N**-oxides. Structural, thermochemical as well as nucleus-independent chemical shift (NICS) criteria are employed to probe the potential aromaticity, antiaromaticity and nonaromaticity of **N**-methylazepine, its 7-azanorcaradiene valence isomer. For the sake of comparison, analogous studies are performed on **N**-methylpyrrole and its **N**-oxide.

## 1. Introduction

Azepine **1** was first synthesized by Hafner [1] and its high reactivity noted. Its 7-azanorcaradiene valence isomer **2** was undetectable by NMR in CDCl_3_ solution at −60 °C [2]. Paquette and coworkers investigated the thermal chemistry of simple **N**-substituted derivatives and found that they dimerized via concerted 6π + 4π cycloaddition [3,4]. Interestingly, they found that the azanorcaradiene **3** could be synthesized while only the azepine structure was observed for **4** [3,4]. Substitution at the **2**, **4**, and **7** positions sterically retards this cycloaddition reaction and stabilizes these azepine derivatives to relatively low-temperature dimerization [3,4]. However, at higher temperatures (>200 °C), a new pathway prevails: ring contraction to the substituted benzene through the intermediacy of trace concentrations of the 7-azanorcaradiene [5].



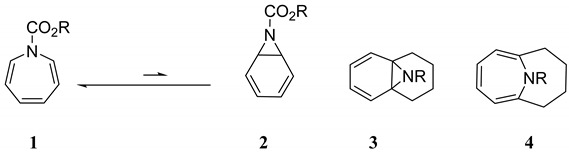



It is worthwhile to note that there are a number of pharmacologically active benzazepines and dibenzazepines including carbamazepine (**5**) as well as depramine and opipramol [6]. Fused benzene rings provide enhanced stability to these azepines.



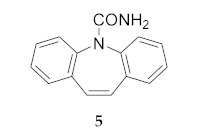



What are the properties of the **N**-oxides of azepines? From a theoretical perspective would **N**-oxidation convert **N**-methylazepine, generally considered mildly antiaromatic [7], to a nonaromatic or even a mildly homoaromatic molecule? Could azepine **N**-oxides exhibit unique chemistries which allow formation of novel azepines when reaction is followed by reduction? What change is there in the energy and equilibrium relationship with the 7-azanorcaradiene valence isomer? Experimental and computational data indicate a bond dissociation enthalpy (BDE) of 62−64 kcal/mol for pyridine **N**-oxide with a value 8−14 kcal/mol lower for trimethylamine **N**-oxide [8]. How do the **N**-O BDE values for azepines and their 7-azanorcaradiene valence isomers relate to the thermochemistry of these molecules? In this regard, **N**-methylpyrrole is an interesting counterpoint to **N**-methylazepine. Since **N**-methylpyrrole has over 20 kcal/mol of aromatic resonance energy (see discussion below), the corresponding **N**-O BDE should be extremely low, casting doubt over whether its oxide could be isolated or even observed at extremely low temperatures. These are among the questions the present paper will explore employing calculations based on density functional theory.

## 2. Results and Discussion

### 2.1. **N**-Methylazepine, **N**-Methyl-7-Azanorcaradiene, and Their **N**-Oxides

Scheme 1 depicts the interconversion pathways between the two stereoisomers of **N**-methylazepine (**6a** and **6b**) and the two stereoisomers of its azanorcaradiene valence isomer (**7a** and **7b**). Simple amine inversion is also a facile interconversion pathway between **6a** and **6b** while **N**-inversion of aziridines (**7a** and **7b**) have much higher barriers [9]. The large difference in enthalpy between **6** and **7** is consistent with the computational and experimental comparisons with CH_2_, O, S, and P analogs of N in the equilibrium between **6** and **7** [10]. Scheme 2 depicts the interconversion pathways between the corresponding **N**-oxides of **6** (**8a** and **8b**) and **7** (**9a** and **9b**). Table 1 lists calculated relative enthalpies and relative free energies of **6a**, **6b**, **7a** and **7b**. Table 2 lists calculated relative enthalpies and relative free energies of **8a**, **8b**, **9a**, and **9b**. What is striking is the significant reversal in the stabilities of the azepine and 7-azanorcaradiene valence isomers (see M06/6-311+G(d,p) results in Table 1 and Table 2). The reversals in enthalpies and free energies can be as much as 10 kcal/mol. The interesting question is what is the fundamental source of these large changes in order: the azepine more stable than its 7-azanorcaradiene and the order reversed for the **N**-oxy valence isomers?

In order to further investigate this dramatic crossover, it is worthwhile to explore a variety of comparisons with other measures of stability including **N**-O BDE values, thermochemical measures of aromaticity as well as calculated structural and nucleus-independent chemical shift (NICS) [11,12,13] data. While the experimental [14] and computational data [8] indicate a BDE for pyridine **N**-oxide (**10**) of 62–64 kcal/mol (eqn 1), the corresponding value for pyrrole **N**-oxide (**11**, eqn 2, see Table 3) is far lower, reflecting resonance energy of ca. 20–25 kcal/mol in the aromatic **N**-methylpyrrole, as described below. Table 3 lists calculated **N**-O BDE values for some other amine oxides. The M06/6-311+G(d,p) BDE values for **N**-methylazepine **N**-oxide, while much higher than that of pyrrole **N**-oxide, are surprisingly low compared to trialkylamine **N**-oxides such as trimethylamine **N**-oxide.


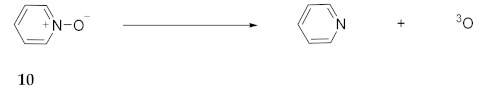
(1)


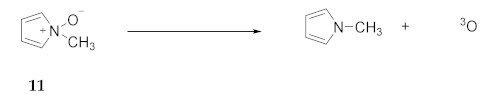
(2)

Figure 1 depicts the enthalpy relationships among **6**–**9**. The largest effects appear to arise from two factors: the relative stability of **N**-methylazepine (**6**) relative to its 7-azanorcaradiene valence isomer (**7**) and the relatively low **N**-O BDE of **N**-methylazepine **N**-oxide (**8**). Both the 7-azanorcaradiene (**7**) and its **N**-oxide (**9**) appear to exhibit no unusual destabilization or stabilization effects on enthalpy. Indeed, subtracting Equation (4) from Equation (3) indicates that the difference between the **N**-O BDE of 9 and the model trimethylamine **N**-oxide (TMAO) is only about 1 kcal/mol (see isodesmic Equations (3) and (4) and Table 4). In contrast, the **N**-O BDE for **N**-methylazepine **N**-oxide (**8**) is about 8 kcal/mol lower than that of its azanorcaradiene valence isomer **9**. This suggests destabilization in 8 that, as noted above, is not present in 9. What is the source of this destabilization? Although it will be further discussed in the next section, it is worthwhile examining isodesmic Equation (5). This equation is endothermic by only 1.9 kcal/mol indicating very similar BDE values in two systems having similar strain energies and therefore similar sources of destabilization.


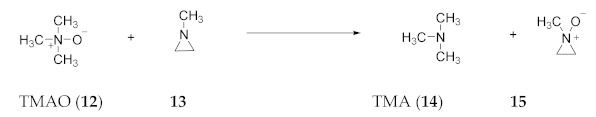
(3)


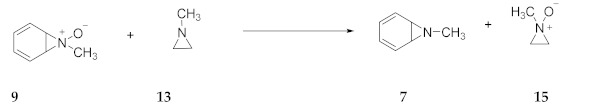
(4)


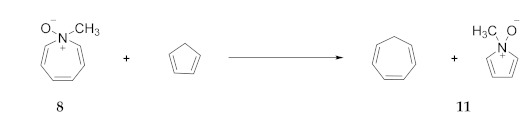
(5)

Are aziridine **N**-oxides such as the simple **15** or the 7-azanorcaradiene **N**-oxide **9** stable or even observable? Although there are published computations suggestive of the intermediacy monocyclic aziridine **N**-oxides [15,16], definitive experimental work was published by Baldwin and coworkers [17]. They observed that ozonolysis of **N**-*tert*-butylaziridine at −75 °C in methylene chloride produced the **N**-oxide **16**. **N**-*tert*-Butylaziridine **N**-oxide **16** was observable by NMR up to 0 °C, above which it decomposed to ethylene and *tert*-nitrosobutane (Equation (6)). Consistent with their earlier studies of episulfoxides, thermolysis of the 2-methyl **N**-oxide **17** at −30 °C produced the hydroxylamine **18** involving H-transfer from the methyl substituent *syn* to oxygen (ene reaction, Equation (7)). Since extrusion Equation (6) yields two molecules from a single molecule, the increase in translational degrees of freedom introduces a significant positive entropy contribution not present in the ene rearrangement (Equation (7)). This pathway has an experimental energy of activation of 15 ± 1 kcal/mol compared to the extrusion reaction (Equation (6)) with an experimental energy barrier of 22 ± 1 kcal/mol [17]. Thus, despite the favorable entropy factor, the extrusion reaction (Equation (6)) is slower than the rearrangement (Equation (7)). For 7-azanorcaradienes (**9a** and **9b**) the ene reaction cannot occur. But the slower extrusion reaction, analogous to (Equation (6)), is calculated to be highly exothermic, −31.2 kcal/mol according to APFD/aug-cc-pVTZ (B3LYP/6-31G**: −42.7 kcal/mol; APFD/aug-cc-pVDZ, −31.7 kcal/mol). The positive entropy (−TΔS) of extrusion is consistent with our calculated free energy some 13 kcal/mol more negative than the calculated enthalpy of extrusion. Thus, the free energy of activation for extrusion to form benzene should be significantly lower than 22 kcal/mol but perhaps high enough to allow observation of 9a and/or 9b at temperatures somewhat below 0 °C but significantly higher than −75 °C.


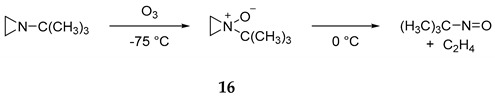
(6)


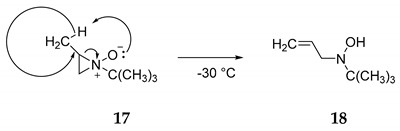
(7)

### 2.2. Aromaticity and Antiaromaticity of Pyrroles, Azepines and Their **N**-Oxides

In attempting to understand the energetics of **N**-methylazepine and its **N**-oxide, it is interesting to think about aromaticity and antiaromaticity. 1*H*-Azepine has been assigned significant antiaromaticity despite its boat-like conformation [7]. **N**-protonated 1*H*-azepine was assigned mild homoaromaticity due in part to the C_2_-C_7_ proximity [7,18].

Aromaticity (antiaromaticity) is a fundamental concept to account for structure and reactivity of electron delocalized cyclic compounds, intermediates and transition states. The earliest postulates applied to cyclic π-delocalized systems (Hückel Rule [19,20,21,22]) but more recently include cyclic delocalized s-systems as well [23]. Aromatic species are characterized by their thermodynamic properties of inherent stabilities [24], structures with hybrid bond lengths [25], specific reactions such as preferred substitution not encountered by acyclic or nonaromatic systems [26], and secondary magnetic fields generated by ring current effects detectable by magnetic resonance spectroscopy (NMR for example). On the other hand, antiaromatic species are destabilized energetically relative to their acyclic models, exhibit unusual kinetic reactivity with structures showing traditional multiple and single bond lengths [27]. In some cases, these destabilized intermediates show ope**N**-shell triplet diradical behavior in their reactivity [28].

Although there are no quantitative methods to measure absolute or even relative aromaticity, certain parameters have been used in order to probe this phenomenon. These include thermodynamic parameters such as heats of formation and enthalpies of hydrogenation, many of which have been determined experimentally employing calorimetry. Other methods include structure analyses and observation of bond length equalization, an indicator of electron delocalization. The theoretical approach developed by Schleyer, the nucleus-independent chemical shift (NICS) [11,12,13], has been widely used to probe aromaticity as well as antiaromaticity and will be further discussed later.

In the current endeavor the question arises as to whether **N**-methylazepine **N**-oxide is aromatic (homoaromatic), or nonaromatic. In order to address this point the vinylogous **N**-methylpyrrole **N**-oxide and its deoxygenated product, **N**-methylpyrrole were examined, since more experimental data are available for the latter which can be crosschecked with the computational results for accuracy of the different methods of calculation. The analyses were carried out using enthalpies of hydrogenation in order to obtain resonance stabilization energies, computed structures to examine hybrid bond lengths, and computed NICS values. It is also useful to refer back to isodesmic Equation (5).

### 2.3. Enthalpies of Hydrogenation and Resonance Stabilization

Enthalpies of hydrogenation were computed for the following reactions (Scheme 3):

The enthalpies were computed at three levels and are listed in Table 5, where APFD/aug-cc-pVTZ is the highest level of theory. No experimental thermochemical data are available for this group of hydrogenation reactions.

A resonance stabilization energy (RSE) for **N**-methylpyrrole can be obtained from the computed data by taking the difference between ΔH_hyd_ for **N**-methylpyrrole and two times that for the corresponding 2,3-dihydropyrrole leading to a resonance stabilization energy (RSE) value of 19.6 kcal/mol by APFD-aug-cc-pVTZ (B3LYP/6-31G(d,p) gives 22.3 kcal/mol; APFD/aug-cc-pVDZ gives 23.2 kcal/mol). This method overestimates the RSE since there is a stabilization when alkenes are conjugated in the model systems. Although no RSE stabilization energy have been reported for **N**-methylpyrrole, thermochemical data for pyrrole indicate an RSE of 24 kcal/mol from combustion heats and 27 kcal/mol using structure group increments [29]. This is abundantly clear through cursory inspection of Table 5, where values in column (1) are quite close those in columns (2) and (3) despite double the uptake of hydrogen.

For **N**-methylpyrrole **N**-oxide and its hypothetical reduction products (Scheme 4) a similar analysis follows and the computed enthalpies of hydrogenation are listed in Table 6:

A resonance stabilization energy (RSE) (or effectively destabilization since the calculated values are negative) can be obtained from the computed data by taking the difference between ΔH_hyd_ for **N**-methylpyrrole **N**-oxide and two times that for the corresponding 2,3-dihydropyrrole **N**-oxide leading to a RSE value of −2.4 kcal/mol by APFD/aug-cc-pVTZ (B3LYP/6-31G(d,p) gives −2.3 kcal/mol; APFD/aug-cc-pVDZ gives −2.3 kcal/mol). Thus, the **N**-oxide appears to be at best nonaromatic (or possibly antiaromatic akin to cyclopentadienyl cation in a planar sp^2^ nitrogen hybridization). The aromaticity of **N**-methylpyrrole and the nonaromaticity for its **N**-oxide are also corroborated by the calculated NICS(0) values as shown in Table 7:

### 2.4. Aromatic Delocalization from Equalization of Double and Single Bond Lengths

For **N**-methylpyrrole experimental structure determinations using two methods, gas phase electron diffraction (GED) and microwave (MW) spectroscopy have been reported [30]. These data (Table 8A) can be compared with the current computational data (Table 8B). As can be seen the computed data for bond lengths at all levels compare favorably with the experimental values attesting to the accuracy of the computation methods. Of note is the equalization of the C-C bonds of bond 2 and 3 from both experiment and theory indicating extensive delocalization.

For the corresponding **N**-oxide, no experimental structure determination has been reported. The computed bond lengths are reported in Table 9. The absence of the lone pair on nitrogen in the **N**-oxide inhibits any cyclic π-delocalization resembling an acyclic 1,3-cyclopentadiene moiety. Summarizing these results, it appears that **N**-methylpyrrole is aromatic but the corresponding **N**-oxide is nonaromatic as would be expected.

### 2.5. Enthalpies of Hydrogenation and Resonance Stabilization of **N**-Methylazepine

According to Hückel’s rule, this cyclic π-configuration possesses eight electrons and would be predicted to be antiaromatic in its planar conformation. In order to avoid such destabilization effects a puckered conformation can be adopted (vide infra). Enthalpies of hydrogenation were computed for the following four reactions and are listed in Table 10. No corresponding experimental thermochemical data are available. Enthalpies of hydrogenation were computed for the following reactions (Scheme 5):

The RSE for **N**-methyl azepine can be obtained from the computed data by taking the difference between ΔH_hyd_ for **N**-methylazepine (1) versus two times that for 2,3,4,5-terahydroazepine (2) plus that for 2,3,6,7-tetrahydroazepine (3) leading to an RSE value of 6.5 kcal/mol by APFD/aug-cc-pVTZ (B3LYP/6-31G(d,p) gives 7.3 kcal/mol; APFD/aug-cc-pVDZ gives 7.5 kcal/mol). This method overestimates the RSE since the model system would have a doubly conjugated (relative to 3) alkene moiety adding an additional 4 kcal/mol in the model system.

For the **N**-methylazepine **N**-oxide and its hypothetical reduction products (Scheme 6) a similar analysis follows and the computed enthalpies of hydrogenation are listed in Table 11. No corresponding experimental thermochemical data are available for this as-yet-unknown molecule.

The RSE for **N**-methyl azepine **N**-oxide can be obtained from the computed data by taking the difference between ΔH_hyd_ for **N**-methylazpine **N**-oxide (1) and two times that for 2,3,4,5-terahydroazepine **N**-oxide (2) plus that for 2,3,6,7-tetrahydroazepine **N**-oxide (3) leading to an RSE value of 13.2 kcal/mol by APFD/aug-cc-pVTZ (B3LYP/6-31G(d,p) gives 15.2 kcal/mol APFD/aug-cc-pVDZ gives 15.4 kcal/mol). Once again, this method overestimates the RSE since the model system would have a doubly conjugated (relative to 3) alkene moiety adding an additional 4 kcal-mol^−1^ in the model system. Nevertheless, **N**-methylazepine **N**-oxide appears to be more thermochemically stable than **N**-methylazepine based on these hydrogenation enthalpies. Could **N**-methylazepine **N**-oxide be slightly homoaromatic?

The aromaticity-antiaromaticity-nonaromaticity of **N**-methylazepine and its **N**-oxide can be probed from the calculated NICS(0) values as shown in Table 12:

This suggests that **N**-methylazepine has measurable antiaromatic character, whereas its **N**-oxide appears to be nonaromatic relative to benzene (+9.83 (B3LYP/6-31G(d,p)); +8.24 (APFD/aug-cc-pVTZ) or **N**-methylpyrrole (+15.15 (B3LYP/6-31G(d,p)); +13.38 (APFD/aug-cc-pVTZ)).

### 2.6. Computed Structures for **N**-Methylazepine and Its **N**-Oxide

Table 13 provides selected experimental structural data for the boat-like **N**-methylazepine. Table 14 lists the corresponding computational data for **N**-methylazepine **N**-oxide.

The above bond-length data demonstrate single- and double-bond alternation resembling a polyene with very minimal equalization. The **N**-oxide appears to exhibit similar alternating single and double C−C bond lengths resembling a conjugated acyclic polyene not showing bond equalization becoming of an aromatic species. The computed bond lengths for both **N**-methylazepine and its **N**-oxide are similar to computed values for the parent azepine [31]. The structural analyses for both derivatives point to their nonaromatic nature at best. The possibility of the **N**-oxide showing homoaromaticity by probing the C2−C7 distance shows, contrary to expectations, a slightly larger value for the oxide. Figure 2A,B compare the lowest energy conformations of **N**-methylazepine and **N**-methylazepine **N**-oxide. Both rings are significantly puckered and are very similar in structure. Figure 2C shows the fully planar form of the **N**-oxide, calculated to be only 1.6 kcal/mol (6.9 kJ/mol) higher in enthalpy than the most stable conformer (Figure 2B). Interestingly, the ring-inversion barrier of some simple **N**-substituted azepines appears to be quite low [4]. Perhaps the degree of aromaticity in the ground state and transition state for ring inversion are fairly similar.

The antiaromatic character of azepines was shown indirectly with azepine **N**-carboxylates from studies of restricted amide bond rotation using variable temperature NMR spectroscopy. It was reported that azepine derivatives showed greater free energy activation barriers compared to 4,5 dihydro derivatives [32]. This study points to the avoidance of the nitrogen lone pair to conjugate with the ring π-electrons preferring delocalization into the carbonyl group.

### 2.7. Experimental Evidence for Azepine **N**-Oxides

There is another interesting point relating to the **N**-alkylazepine **N**-oxides. In an experimental study of the oxidation by *m*-chloroperbenzoic acid (*m*-CPBA) of **N**-substituted-dibenz[*b*,*f*]azepines (**19**), the 10,11-epoxide (**20**) was formed, at least initially, when R = *n*-C_3_H_7_, CH_2_C_6_H_5_, or CH_2_CH_2_CH_2_OH. The epoxide is also formed when R = COCH_3_ (carbamazepine, 5) consistent with the report of the 10,11-epoxide as a urinary metabolite of carbamazepine [33]. These results are consistent with the study of dibenz[*b*,*f*]azepines using rat liver microsomes [34] as well as studies modelling oxidative metabolism using Fenton reagent (Fe(II); H_2_O_2_) [35]. Fenton reagent also epoxidized the parent compound (**19**, R = H) to form **20** (R = H) [35]. Interestingly, only the **N**-methyl compound (**19**, R = CH_3_) formed an **N**-oxide (**21**) with *m*-CPBA [36]. The amide linkage in carbamazepine makes this and related azepine **N**-amides resistant to **N**-oxide formation. Steric hindrance in the vicinity of nitrogen also plays a role [36]. However, the observation that the parent compound (**19**, R = H) epoxidizes, while the corresponding methyl compound forms **21** suggests that electro**N**-donation by methyl may also play a role. Clearly, **21** (mp 151–155 °C, decomp.) [36] will not isomerize to its far-less-stable, highly strained and nonaromatic azanorcaradiene valence isomer. If this isomer were accessible, it would readily form phenanthrene upon extrusion of nitrosomethane.

Comparison of the calculated enthalpies of **N**-methylazepine **N**-oxide (**8**) with the 4,5-epoxide isomer **22** indicates that the epoxide is about 45–46 kcal/mol (B3LYP/6-31G(d); M06/6-311+G(d,p) respectively) more stable than the **N**-oxide (the 2,3 epoxide is calculated to be 1–2 kcal/mol lower in enthalpy than **22**). This very large energy difference may come as a surprise at first. However, if one compares the enthalpy of expulsion of ^3^O from *trans*-1,2-dimethylethylene oxide (+88 kcal/mol) with the BDE of even a relatively strong N−O bond (61.5 kcal/mol, pyridine **N**-oxide, Table 3), epoxidation is favored by about 26 kcal/mol [8,37]. Comparison with the considerably weaker N−O bond in **8** (BDE ca 41–42 kcal/mol, Table 3) predicts epoxidation to form **22** is favored by about 47 kcal/mol. This does suggest that the cited addition of oxidation to form **21** is fully under kinetic control and that the **N**-methyl substituent confers both optimal steric and electronic factors favoring selective **N**-oxidation.



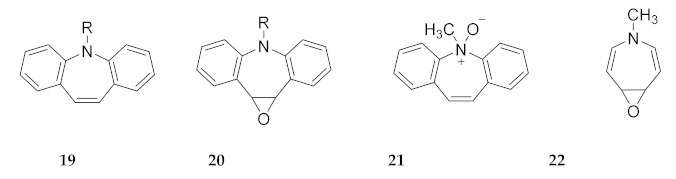



## 3. Materials and Methods

Energies, zero-point energies, enthalpies and free energies were calculated with the GAUSSIAN 09 [38] and GAUSSIAN 16 [39] software packages. From these we obtained bond dissociation enthalpies (BDE). The structures were optimized, and vibrational frequencies calculated, with standard methods. The calculated frequencies were all positive: all structures reported here were minima of energy surfaces. Calculations were based on density function theory and different exchange-correlation functionals: B3LYP [40], APFD [41], and Minnesota M06 [42]. We used these basis sets: Pople’s 6-31G(d), 6-31G(d,p), 6-311+G(d,p) [43,44] and Dunning’s aug-cc-pVDZ [44] and aug-cc-pVTZ [45] basis sets. Our biggest basis set, aug-cc-pVTZ, contains polarization functions and diffuse functions of s, p, d, and f-type. Where we showed results with different methods, the highest levels of theory were preferred: first, APFD/aug-cc-pVTZ; second, M06/6-311+G(d,p). The nucleus-independent chemical shifts (NICS(0)) were calculated using the NMR shielding tensors and magnetic susceptibilities [46] module with the gauge-independent atomic orbital (GIAO) [47,48] approach by B3LYP. Enthalpies and free energies were calculated at 298.15 K and 1 atm.

## 4. Conclusions

Oxidation of amines to amine **N**-oxides, using hydrogen peroxide, ozone, *meta*-chloroperbenzoic acid among other reagents, produces interesting changes in structure and stability. The **N**-methylazepine system, mildly antiaromatic by structural, energetic and NICS(0) criteria in the Appendix A, should form an interesting **N**-oxide. Whereas **N**-methylazepine is calculated to be 4–8 kcal/mol (17–33 kJ/mol) lower in enthalpy than its 7-azanorcaradiene valence isomer, the 7-azanorcaradiene **N**-oxide is calculated to be about 1–5 kcal/mol (4–20 kJ/mol) lower in enthalpy than **N**-methylazepine **N**-oxide. The sources of this roughly 8 kcal/mol (33 kJ/mol) reversal (See Figure 1) in isomer stabilities are not obvious. **N**-methylazepine **N**-oxide appears to be nonaromatic, rather than antiaromatic, using structural, energetic and NICS(0) criteria. The 7-azanorcaradiene **N**-oxide appears to be no more stabilized or destabilized than **N**-methylaziridine **N**-oxide or even trimethylamine **N**-oxide (TMAO) according to isodesmic equations. A most striking example is the presently unknown **N**-methylpyrrole **N**-oxide. Its N−O BDE is calculated to be only 13–18 kcal/mol (54–75 kJ/mol) in stark contrast to pyridine **N**-oxide (60−64 kcal/mol or 250–270 kJ/mol). It seems unlikely that this amine **N**-oxide will ever be isolated or even observed spectroscopically. Dissociation of the N−O bond in **N**-methylpyrrole **N**-oxide yields aromatic **N**-methylpyrrole with RSE of 24−27 kcal/mol (100–110 kJ/mol). But added to this is the intrinsic ca. 8 kcal/mol (33 kJ/mol) destabilization in the amine oxide (N^+^-O^−^) linkage flanked by two vinyl groups. This appears to be the source of the 8 kcal/mol destabilization in **N**-methylazepine **N**-oxide (**8**) which causes it to be less stable than its azepine **N**-oxide isomer **9** and reduces its N−O BDE by about 8 kcal/mol compared to its isomer as vinyl groups are destabilized by electro**N**-withdrawing substituents [49,50]. While a very thorough computational study on the effects of substituents on the 1H-azepin/-7-azanorcaradiene system has been published [51], the study focuses on substitution at the 3- and 4-positions of azepine but not on **N**-substitution. Extrusion of tert-nitrosobutane from **N**-*tert*-butylaziridine **N**-oxide occurs at 0 °C in dichloromethane solution with an energy of activation of 22 kcal/mol. The same extrusion reaction from **N**-*tert*-butyl-7-azanorcaradiene-**N**-oxide will be considerably more exothermic, have a significantly lower energy of activation but the **N**-oxide, generated from the amine at −75 °C, is likely to be observable at higher temperatures although significantly below 0 °C.

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
