# Peer review of "Predicted Reversal in N-Methylazepine/N-Methyl-7-azanorcaradiene Equilibrium upon Formation of Their N-Oxides"

_molecules, 2020, doi:10.3390/molecules25204767_

Round 1
Reviewer 1 Report
The manuscript presents na extensive analysis on the structural, enthalpy and Gibbs energy effects involving the conversion of the azepine to the corresponding azepine N-oxide. The authors used adequate computational tools and theoretical approaches to evaluate geometry optimizations and to obtain frequency calculations in order to derive bond dissociation energies, bond dissociation enthalpies, enthalpies and Gibbs energies.
The manuscript is well organized, well written and there is a very good bibliographic support to the study. The authors are well aware of this type of theoretical studies and this gives them the willingness to apply the methods involved in the evaluation of the properties of the systems under study.
In my opinion, the manuscript may be accepted without changes.
There is one detail that I regret has not been taken into account by the authors: SI units are not used, which makes difficult to the reader the comparison of the values described here with reference data from the literature, leading to the need for conversions that could be avoided.
Reviewer 2 Report
Fournier et al. report a computational study to investigate the fundamental causes of the chemical equilibrium in N-methylazepine and its change upon N-oxidation. The work has potential to be interesting for the organic chemistry and physical organic chemistry communities. However, in its current form, the work contains important methodological defficiencies that in the opinion of this reviewer can be fixed. The work also contains excess of superfluous, or at least unjustified, information while it lacks important discussion of other points. For these reasons, I suggest major revisions prior to reconsideration.
Major points:
1. Metholody 1. Did the authors check that all the vibrational frequencies of the obtained minima were positive for all the molecules to verify that they were obtaining true energy minima? I assume that the authors verified this fundamental point during their research. However, this should be clearly stated in the methods section.
2. Methodology 2. The Basis sets 6-31G* and 6-31G** employed in combination with the B3LYP functional are very small for the size of the computed molecules and current computational power. Also, they lack diffuse functions, which should be a must when investigating anions, or atoms that are suspicious of bearing a significant amount of negative charge like the oxygen atom of the N-oxides or the pi electrons of double bonds, conjugated or aromatic systems. Since there are not hydrogen bonds, it is much more important to add diffuse functions than adding second polarization functions. The smallest basis set to build upon would be something like 6-31+G(d).
3. Methodology 3. What is the purpose of employing increasing basis sets if their effect is not (minimally) discussed? In the current form of the manuscript, why not reporting the data with just one combination of functional and basis set? A proper discussion on this issue is necessary.
4. Methodology 4: Why is B3LYP employed? The authors seem to be aware of the importance of dispersion interactions to account for energy differences in the present molecules as they used the M06 and APFD functionals, which in their own way account for such interactions. If the authors wanted to investigate the effect of changing the functional, the same basis sets should have been used for all three functionals, which is not the case. A proper discussion on this issue is necessary.
5. Methodology 5. What is the purpose of employing different density functionals if their effect is not discussed? A proper discussion on this issue is necessary.
6. Methodology 6. The data employed to explain the nature of the energy differences (e.g. Figure 1) is B3LYP with the smallest basis sets rather than the better functionals with larger basis sets. Why?
7. Methodology 7. The authors are encouraged to to perform non-DFT calculations such as MP2 or CCSD(T), or at least MP2//CCSD(T) single point calculations to assess the quality of their DFT data. Optionally, they are encouraged to find, cite and properly discuss their choice of DFT method based on previous published research. Given the size of the systems, performing these calculations should be feasible and it would enrich the investigation.
8. The investigation seems to be incomplete. Why are the enthalpies and free energies of some molecules not shown in Tables 1 and 2? (e.g. those of 6b, 7b, 8a, 8b, 9b).
9. In the abstract it is said that up to five different functionals and basis sets are employed but these are only used (incompletely) in tables 1-3. Why are the rest of the tables lacking data of some functionals and basis sets.
10. What is the rationale behind the statement “The positive entropy of extrusion is consistent with calculated free energies some 13 kcal/mol even more exergonic.” of L117. The entropy change is independent of the enthalpy change, thus any value of entropy change could be consistent with any value of free energy change. In this particular case, the entropy changes for the extrusion reactions should clearly be independent of the enthalpy of the reaction (mostly dependent on the nature of the broken/formed chemical bonds) since most of the entropy change should arise from the gain of traslational and rotational degrees of freedom of the dissociated molecules. Please explain this statement.
11. It is not clear why reactions 3 to 6 are calculated and discussed. The authors indicate that it is worthwhile to investigate the change in isomerization energies of molecules 6-9 but they do not draw any explanation of the data in Tables 1 and 2 from the data in Tables 3 and 4.
12. What is the purpose of the data about the aromaticity of N-methylpyrrole and its N-oxide (sections 2.3 and 2.4)? What is its contribution to explain the chemistry of N-Methylazepine? Sections 2.3 and 2.4 are excessive to explain the data reported in section 2.5.
13. The conclusion in L243 "Nevertheless, N-methylazepine N-oxide appears to be more thermochemically stable than N-methylazepine based on these hydrogenation enthalpies." is wrong. The relative stability of N-methylazepine relative to its N-oxide should be calculated as the authors do in eqs. 1 and 2 and shown in Fig. 1. In section 2.5, the comparison is made between two estimations of the resonance stabilization energy, which by itself does not account of the total thermodynamic stability of one molecule respect to another as other factors such as the formation of the N-O bond, its resulting electron reorganization, and all the interactions between the O atom with the other atoms also play a role that is not considered in the author's comparison.
14. Important answers are missing. It is shown that N-oxidation reverses the equilibrium between the N-Methylazepine and 7-azanorcaradiene. It is shown that the BDEs are larger for the 7-azanorcaradiene isomers. Also that the N-methylazepine is antiaromatic while its N-oxide is non-aromatic. However, we don't know a clear answer to the question posed in L70 "The interesting question is what is the fundamental source of these large changes in order: the azepine more stable than its 7-azanorcaradiene and the order reversed for the N-oxy valence isomers?"
15. Something that can indeed be more interesting is to provide data to confirm or deny the question in L292 "Perhaps the degree of aromaticity in the ground state and transition state for ring inversion are fairly similar." This should be easy to check by finding such transition state and performing the corresponding aromaticity analysis.
16. The authors use the term "normal" several times. Maybe the use of this language is fine for an informal oral conversation but not for a written communication in a journal. What is it that is "normal"? In which terms? By which measure? Please, explain the same with objective and precise language.
Minor points:
17. Lines 254-257, the formats of the introduction to Table 13 and the Table title are exchanged.
18. The label “10” of pyridine N-oxide in eq. 1 is incorrectly assigned to pyridine rather than to the oxide.
19. In eq. 4 the label “8” should be “9”.
20. Add the molecule labels to Figure 1
21. Explain the acronym BDE the first time it is used (L50), currently in L334
22. Statements such as “Since N-methylpyrrole has over 20 kcal/mol of aromatic resonance energy” or “Simple amine inversion is also a facile interconversion pathway between 6a and 6b while N-inversion of aziridines (7a and 7b) have much higher barriers.” require proper citation.
Round 2
Reviewer 2 Report
To a greater or lesser extent, the authors have addressed many of the comments made by this reviewer. However, there are some important points that were not addressed or not satisfactorily.
1) Point 13 of the first round of review. The authors agree that hydrogentation enthalpies do not necessarily correlate with thermodynamic stabilities. They have the data to calculate properly the thermodynamic stability and validate their assumption, yet they prefer not to do it. I do not understand why.
3. Show all the relative energies in tables 1, 2, 3. The authors keep using terms as small or large without providing exact quantities based on their calculations. The manuscript must explicitly include the quantities that lead the author's to draw their interpretation so that the reader can agree (or disagree) with their interpretation. Again, the authors already have the data, the cost of reporting it is zero.
2. The sentence (L130) regarding the entropy of extrusion is still confusing. What do the authors mean by an entropy being consistent with a free energy? Please, explain clearly. In addition, they compare an entropy increment with a free energy in terms of kcal/mol. However, entropies are expressed in terms of energy/temperature (i.e. in this case kcal/mol·K.) Are the authors reporting the product of temperature and entropy increment (TDeltaS)? If so, also state it clearly.
